# Significant increase of global anomalous moisture uptake feeding landfalling Atmospheric Rivers

Iago Algarra [1], Raquel Nieto [1], Alexandre M. Ramos [2], Jorge Eiras-Barca[1,3], Ricardo M. Trigo [2,4] & Luis Gimeno [1✉]

One of the most robust signals of climate change is the relentless rise in global mean surface temperature, which is linked closely with the water-holding capacity of the atmosphere. A more humid atmosphere will lead to enhanced moisture transport due to, among other factors, an intensification of atmospheric rivers (ARs) activity, which are an important mechanism of moisture advection from subtropical to extra-tropical regions. Here we show an enhanced evapotranspiration rates in association with landfalling atmospheric river events. These anomalous moisture uptake (AMU) locations are identified on a global scale. The interannual variability of AMU displays a significant increase over the period 1980-2017, close to the Clausius-Clapeyron (CC) scaling, at 7 % per degree of surface temperature rise. These findings are consistent with an intensification of AR predicted by future projections. Our results also reveal generalized significant increases in AMU at the regional scale and an asymmetric supply of oceanic moisture, in which the maximum values are located over the region known as the Western Hemisphere Warm Pool (WHWP) centred on the Gulf of Mexico and the Caribbean Sea.

[1] Environmental Physics Laboratory (EPhysLab), CIM-UVIGO, Universidade de Vigo, 32004 Ourense, Spain. [2] Instituto Dom Luiz (IDL), Facultade de Ciências, Universidade de Lisboa, 1749-016 Lisboa, Portugal. [3] Department of Atmospheric Sciences, University of Illinois at Urbana-Champaign, Urbana-Champaign, IL, USA. [4] Departamento de Meteorologia, Instituto de Geociências, Universidade Federal do Rio de Janeiro, Rio de Janeiro 21941-916, Brazil. ✉email: l.gimeno@uvigo.es

Atmospheric Rivers (ARs) are narrow regions through which large amounts of moisture are transported towards midlatitudes (e.g., [1]) usually associated with the pre-frontal region of extratropical cyclones. These structures are responsible for the horizontal transport of large quantities of water vapor, in up to 3–5 different events per hemisphere at any one time, from the subtropics to the mid- and northern latitudes along a relatively narrow (< 1000 km) and elongated (> 2000 km) atmospheric pathway at lower atmospheric levels[2,3]. At a synoptic time scale, ARs are generally associated with low-level moisture convergence located in the warm sectors of extra-tropical cyclones, ahead of pre-cold fronts[4]. Most of extreme precipitation and flood events are associated with landfalling AR events for certain coastal mid-latitude regions, especially when subjected to orographic lift over mountainous topography[5]. They have been linked to a wide range of socio-economic impacts, affecting the severity and frequency of flooding, and occasionally defining the end of a drought period. Additionally, it has been shown that the lack of ARs is often correlated with drought periods (e.g., [6]). Their importance in extreme precipitation events and floods has been analyzed in some detail, mostly for the Western U.S., where ARs are the primary driver of damage caused by flooding[7], and also in western Europe, where intense rainfall is strongly associated with ARs, their persistence being especially relevant as the precursors of winter floods[8–11].

ARs act as bridges or conveyors between oceanic evaporation and continental precipitation[12,13], the former being the main origin of water vapor that reaches land masses at extratropical latitudes. ARs have also been linked with changes in water storage and the mass balance of ice sheets, glaciers, and snowpack[14,15]. The current relentless rise in global mean surface temperature is closely linked to the increase in atmospheric water vapor[16,17], and the greater availability of water vapor favors a larger transport of moisture, and hence an intensification of extreme precipitation events and floods triggered by ARs[18–20]. The water-holding capacity of the atmosphere increases about 7% per kelvin at lower troposphere and for the column integrated moisture, which is mostly concentrated in the lower troposphere[21,22], leading to an intensification of extreme precipitation events at similar rates. We cannot disregard that under climate change, ARs may be affected by changes in dynamics that could alter the strength of the winds (e.g., [23]) or increase the anticyclone activity (e.g., [24]). However, it is generally accepted that thermodynamically driven component dominates[25,26]. In this context, as the amount of moisture in the atmosphere increases, so does the moisture transport. Therefore, weak ARs are bound to grow substantially in future warmer climates achieving more often the magnitude of extreme events, with a potential for a greater impact for humans ecosystems and build-up areas. Simulations of climate warming have shown more intense and frequent ARs[19,25,27–30], which could lead to higher total rainfall and flooding in mid-latitude land masses. Nevertheless, it remains unclear which areas provide anomalous moisture to the ARs, and whether these show trends linked with global warming. In this study, we performed a global assessment to identify those areas where landfalling ARs (hereafter LARs) receive anomalous moisture, and tried to quantify these effects at regional and global scales in terms of the trends in this supply of moisture.

This study confirms the presence of enhanced evapotranspiration rates since the early 1980s in association with LAR events at both regional and global scales. Additionally, we provide evidence of a significant increase of the AMU values, close to the Clausius-Clapeyron (CC) scaling. Overall these results highlight the importance of further intensification of AR predicted by future projections.

## Results

**Global landfall of Atmospheric Rivers.** For the worldwide coastline, and for the period 1980–2017, we first identified the areas of maximum occurrence of LARs (Fig. 1) as those in which the number of ARs detections exceeds the 10% of total days in the time period considered (see Methods). According to the definition, ARs are synoptic systems located in warm sectors of extra-tropical cyclones[4], hence we must be careful in selecting regions of maximum occurrence that lack the appropriate structural fingerprint of ARs. This caveat is particularly relevant for tropical areas influenced by monsoon circulation and regions that do not show a relatively deep negative anomaly of mean sea level pressure (MSLP). Thus, initially 24 regions of maximum occurrence of LARs were identified, but only those regions that showed an associated negative wintertime MSLP anomaly (Supplementary Figs. 1–3), were considered for moisture uptake analysis. A total of four regions were thus excluded from the study (indicated in red in Fig. 1), and our focus was then on the remaining 20 coastal domains.

It is important to highlight the requirement of the availability of water vapor in the atmosphere for moisture transport by ARs. Moisture must have evaporated and accumulated in certain areas before its uptake by an AR. The anomalous moisture uptake (herein denoted AMU) for each region over the LARs was identified as those areas where moisture uptake was intensified during each LAR event using the Lagrangian model FLEXPART v9.0[31,32] forced by ERA-Interim reanalysis[33]. The AMU is defined as the mean anomaly of evaporation along the entire period (1980–2017, see Methods). Figure 2 shows the 90th percentile of AMU areas for each of the retained 20 regions of maximum occurrence of LAR previously identified in Fig. 1. The full values of AMU for each region are given in Supplementary Figs. 1–3. The AMU regions for LARs are mostly located over large upwind oceanic areas, these positions being highly stable in time (Supplementary Figs. 4 and 5 show the weight centroids for the AMU value for each LAR and region). Most regions show a significant increase in AMU for the period 1980–2017, as indicated by the filled areas in Fig. 2, while unfilled areas are those with no significant trends (see individual maps in Supplementary Figs. 6 and 7). It is necessary to emphasize that the region of the Western U.S. does not show a significant trend, which is in line with the work of 34, 20 where no increase in IVT or precipitation trends have been found over a large period of time (Supplementary Fig. 6b, c).

**Global anomalous moisture uptake by landfalling Atmospheric Rivers.** We also investigated AMU by LARs at a global scale. Figure 3a shows AMU regions for all LAR events. In general, higher AMU values are found over subtropical oceanic areas, between 25° and 40° in both Hemispheres. Nevertheless, a maximum AMU value for the LARs is seen in the so-called Western Hemisphere Warm Pool (WHWP), a large evaporative region that includes the Gulf of Mexico and the Caribbean Sea, where high sea surface temperatures favor a higher rate of evaporation, providing moisture to the atmosphere and favoring its subsequent transport to other remote regions (e.g., [35]). The outstanding magnitude of this large AMU maximum results from the oversized importance of the ARs in the northern Atlantic basin when compared to all the other major oceanic basins (Fig. 2) and the asymmetric role of ARs in the NH vs SH (Fig. 3b). In fact, this region acts as a large moisture reservoir for ARs that develop in the North Atlantic Ocean and make landfall towards the northern coasts of the West Atlantic basin (American East Coast, Fig. 2a and Supplementary Fig. 1g, h), Greenland and Iceland (Fig. 2a and Supplementary Fig. 2a, b) and the west coast of Europe (Fig. 2b and Supplementary Fig. 2c–e). To

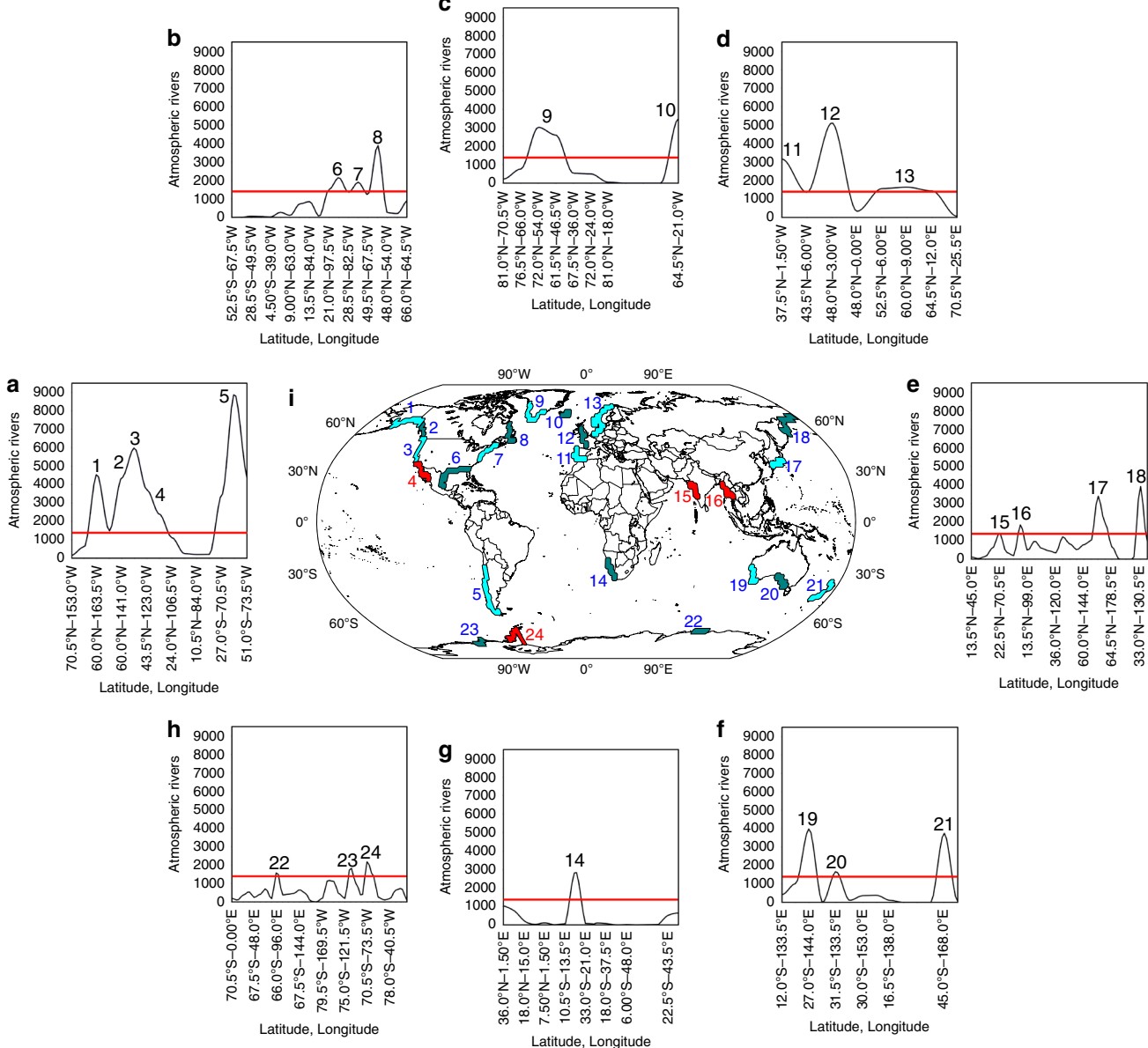

**Fig. 1 Regions of maximum occurrence of landfalling Atmospheric River.** Atmospheric Rivers (ARs) frequency over global coastal areas for the period 1980–2017 for: American West Coast (**a**), American East Coast (**b**), Greenland and Iceland (**c**), European West Coast (**d**), Asian Coast (**e**), Australian Coast and New Zealand (**f**), African Coast (**g**), and Antarctic Coast (**h**). The black line shows the number of ARs against coastal longitude and latitude. The red line represents the threshold of 10% of days computed considering all regions. **i** Global map showing the 24 landfalling Atmospheric River (LAR) regions detected (those not used for the remaining if the study of anomalous moisture uptake are shown in red).

a lesser extent, a local input of AMU closer to the landfalling area of ARs is also seen in the mid-latitudes (Supplementary Figs. 1–3) due to the local convergence of water vapor along the pathways of ARs[36]. Latitudinal variability of AMU is shown in Fig. 3b, and this follows a bimodal distribution. Higher values of AMU are located around 40°N in both hemispheres, and these are twice as high in the Northern Hemisphere, in which the maximum occurrence of LAR was identified. As observed by the shaded gray area in Fig. 3b, the interannual variability is rather small implying that there is scarcely any latitudinal variation in the AMU regions from year to year. Nevertheless, by analyzing the global interannual variation of AMU (Fig. 3c), a significant increase in AMU can be seen for the period studied, of about 0.9% per decade, which is in accordance with the occurrence of significant increases in AMU for each of the 20 regions of maximum occurrence (trends shown in Supplementary Figs. 6 and 7).

## Discussion

In a warmer climate, ARs are expected to become increasingly intense and to increase in frequency[19,29,37], likely escalating their socioeconomic impact[38]. Although it is not known how the moisture transported by ARs will change, the projected increase of 30–40% in the vertically integrated water vapor transport (IVT) in the storm tracks of the Pacific and North Atlantic[18], together with the fact that 9 out of 10 liters of the water vapor that reach extra-tropical latitudes do so via ARs[3], points to an increase in the amount of moisture transport as the temperature rises[24]. The identification and analysis of the variability and long-term changes of the regions that provide moisture for these systems has thus become an essential topic for study.

Our results show the importance of the WHWP region in the AMU for the ARs that develop in the North Atlantic basin. The AMU from the North Atlantic accounts for almost half of the

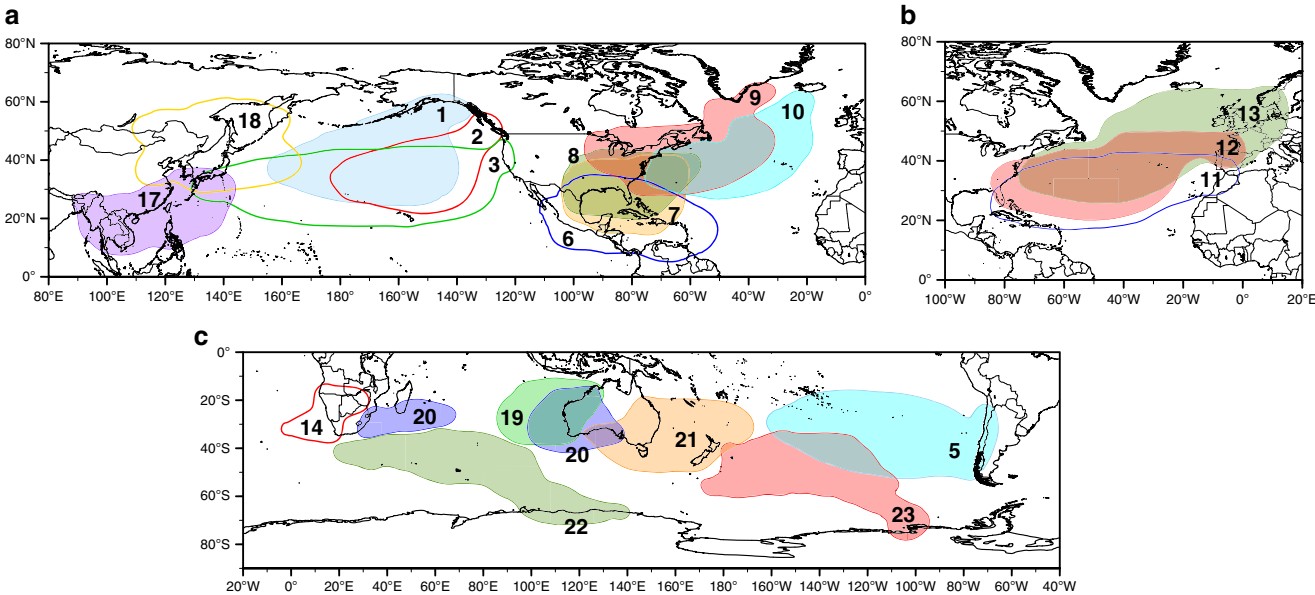

**Fig. 2 Anomalous moisture uptake for regions of maximum occurrence of landfalling Atmospheric Rivers shown in Fig. 1.** 90th percentile of the sum of anomalies of evaporation for each region identified in Fig. 1 (**a**) for north Pacific and North Atlantic Oceans, (**b**) for western Europe and (**c**) for the Southern Hemisphere. Filled areas show a significant increase in evaporation anomalies, while unfilled areas show no significant trend (95% level of significance). The period of study is 1980–2017. Trends were calculated using Mann-Kendall test.

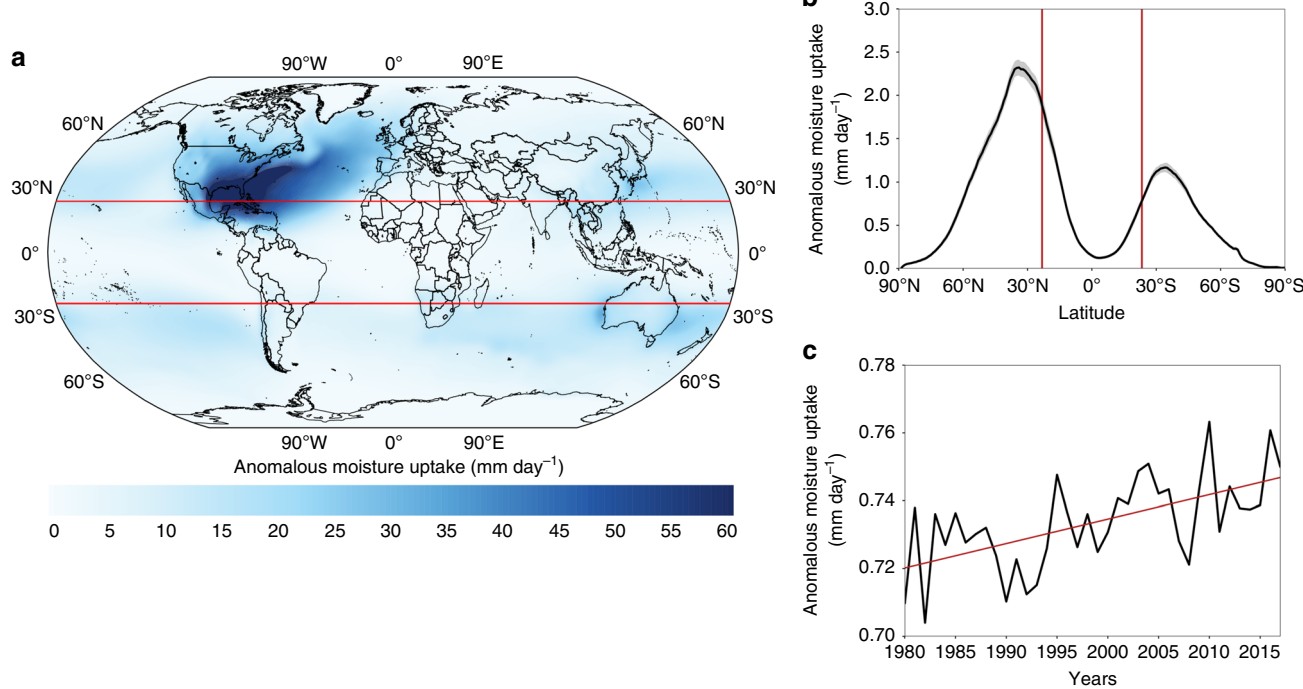

**Fig. 3 Anomalous moisture uptake of the landfalling Atmospheric Rivers events from 1980–2017. a** Annual sum of positive anomalies of moisture uptake for Atmospheric Rivers (ARs) detections over the main landfalling regions. The red lines define the region between the Tropics of Cancer (23.43°N) and Capricorn (23.43°S). The anomalies are calculated over the period 1980–2017. **b** Zonal average of Anomalous Moisture Uptake (AMU, black line) and its interannual variability (shaded gray area). **c** Global annual variability of AMU. The red line shows the corresponding linear regression.

global AMU (Supplementary Fig. 8a–c). When we analyze the annual variability either solely from WHWP or excluding this region from the global analysis, the AMU trend continues to increase significantly (Mann-Kendall test) (Supplementary Fig. 8b–d). Our results are in line with the changes in the sea surface temperature (SST) of the North Atlantic basin, which have revealed a warming trend associated with the North Atlantic

mutildecadal variability (AMV). A recent study[39] shows how anthropogenic emissions of aerosols, particularly sulphate, may be responsible for more than half of the decadal SST variability associated with AMV.

We show a significant increase in the global AMU for landfalling AR events (LARs) in the current climate (about 0.9% per decade for 1980–2017, Fig. 3c). We estimate that extreme

precipitation events will increase at the same rate as atmospheric moisture following the Clausius-Clapeyron (CC) ratio, in which the overall average amount of water vapor increases at a rate of 7.3%/K with respect to the average global surface air temperature[17,40]. Given the critical role of LARs for rainfall and flooding in many regions, their presence may be associated with extreme precipitation, particularly when uplift, either orographic or thermodynamic (via warm conveyor belt), play a significant role[8–11]. Although a large fraction of the top 20 extreme precipitation events are indeed connected with ARs in regions such as the Iberian Peninsula, it is also necessary to stress that many of the detected ARs do not cause extreme precipitation[11]. Therefore, only a fraction of AR events lead to extremes that extremes are also related to non-AR events. Given that the average warming of the global surface temperature for the period 1980–2016 is closer to 0.5 °C[41], our results show a significant increase in AMU in the source regions of about 3% over the same period, or about 6% for each degree of temperature increase, following a relationship close to what is expected from the Clausius-Clapeyron ratio. Analogously to the calculation of AMU, we evaluate the regional and global trend of IVT, which is similar to the trend of AMU in some regions of maximum activity of ARs and shows a significant increase on a global scale (Supplementary Fig. 9).

The relative contribution between the local convergence and tropical advected components of the moisture is still a matter of discussion. Some authors claimed the predominant role played by the former[42,43]. At the same time, other studies found that strong large-scale advection from (sub)tropics had the most important contribution in a variety of case studies both in the US and European West Coast, and both using Lagrangian and Eulerian approaches (e.g., [44,45]). The results stated here confirm that both phenomena are essential in terms of contribution. Well defined AMU regions are located both in (sub)tropical regions and in oceanic regions near to the location where most ARs make landfall. This is in agreement with the conceptual scheme that explains the ARs as large-scale transport structures that are also capable of incorporating substantial amounts of local moisture as they move northward[36].

Despite the robustness of results, it is necessary to quantify the relative roles of the main drivers of evaporation, (SST), near-surface wind speed, and near-surface atmospheric specific humidity[12] in the anomalous moisture transported by ARs. This will constitute a major challenge in future programs of research.

## Methods

**Global identification of AR impact regions**. In order to detect the main regions of LAR occurrence from 1980 to 2017 over the world's coastlines, we use the AR database developed by[46]. This database uses the landfall location for ARs at a global scale, at a spatial resolution of 1.5° and a 6 h time step (00.00; 06.00; 12:00; and 18.00 UTC), based on a threshold of IVT intensity, to which geometric conditions are added according to the coherence of AR structures using ERA-Interim reanalysis data from the European Centre for Medium-Range Weather Forecast (ECMWF)[33]. Further details of the AR detection can be found in[46].

For the present study, LAR events were counted at a spatial resolution of 12° along each coast. Using this new coarser resolution, we identified those points with a frequency greater than 10% of the total number of days with LARs, i.e., 1388 days, considering these to be the regions of maximum LAR occurrence. To assess whether a LAR event was associated with cyclogenesis, we evaluated the regions identified in terms of wintertime mean sea level pressure (MSLP) anomaly (data obtained from ERA-Interim Reanalysis). Those regions that were not associated with negative anomalies were dismissed.

Given the broad region of occurrence of LARs along the west coast of North America, and in order to obtain a comprehensive assessment in terms of AMU, we considered four different sub-regions (areas numbered by 1, 2, 3, and 4 in Fig. 1). This division is somewhat subjective but necessary in order to avoid dealing with ARs reaching California in the same context as those striking British Colombia or even Alaska. Previous studies have classified the trajectories of ARs reaching the West Coast of North America when they penetrate inland in four specific regimes[47,48], this classification being particularly relevant when assessing the role of

ARs in terms of precipitation. Nevertheless, in order to be as objective as possible in this regard and focusing on the study of AMU during LAR events, it seems appropriate to establish a regional subdivision in terms of the frequency of LAR events taking the coastal geographical orientation into account. It is important to note that the occurrence of LAR ranges broadly latitude. Starting at the highest latitude, the first region extends along the southern coast of Alaska, the second and the third extend from 60°N to 52°N (based on LAR events with a northwest orientation), then from 36°N to 52°N (the region of maximum LAR activity)[49,50], with the fourth and final region extending south of 36°N (this is not considered in the analysis of moisture anomaly because it does not show a seasonal pattern of negative MSLP anomalies).

Nevertheless, the Southern California coast is a very active region in terms of strong LARs, which are more closely associated with southern region 3 than LARs in southern Baja California. In fact, the strongest ARs in Northern California are mostly observed in December[34]. These ARs are associated with the Southern California and to some extent with Northern Baja California LARs, while Southern Baja displays a seasonality and a stronger marked difference, partly also due to the influence of the North American monsoon circulation.

**FLEXPART model: anomalous moisture uptake (AMU)**. The global FLEXPART model (FLEXible PARTicle dispersion model) version 9.0[32] was used to quantify the humidity during AR events. FLEXPART[31,51] is a Lagrangian model that enables us to track atmospheric moisture along each trajectory by following air parcels. The model divides the atmosphere homogeneously into approximately 2.0 million air parcels of constant mass, which are advected by a 3-D wind field. In our simulation, FLEXPART uses ERA-Interim reanalysis data from the ECMWF[33], which are available at a 6-h time step (00.00, 06.00, 12.00; and 18.00 UTC) from 1980 to 2017, at a 1° horizontal resolution on 61 vertical levels, from 1000 to 0.1 hPa with approximately 14 model pressure levels below 1500 m and 23 below 5000 m.

To estimate the sources of moisture for AR events, air parcels are tracked backward in time. Thus, changes in the specific humidity along each trajectory by each air parcel can be expressed as (Eq. (1)):

$$e - p = m\frac{dq}{dt} \qquad (1)$$

where ($e - p$) is the evaporation-minus-precipitation budget, which is the freshwater flux of the particle, $m$ represents the mass of each individual air parcel (expressed in kg), $q$ represents the specific humidity (measured in g/kg), and $t$ is time. The diagnosis of the surface freshwater flux is calculated by integrating ($e - p$) over the entire atmospheric vertical column for all the resident air parcels (Eq. 2):

$$(E - P) \approx \frac{\sum_{k=1}^{k}(e - p)}{A} \qquad (2)$$

where ($E - P$) represents the surface freshwater flux, $K$ is the number of air parcels in residence over a specific area, $A$. The backward analysis in time is used to distinguish the origin of the atmospheric moisture in the air masses during LAR events. Thus, a source of moisture can be defined as that region in which the evaporation exceeds the precipitation i.e., ($E - P$) > 0, and the net moisture budget of the tracked air parcels favors the evaporation from the environment to the particles, where there is a positive contribution of moisture.

The moisture residence time is also a key factor to be considered in transport analysis. It also expected to increase in a warmer future[52]. Longer residence times imply the transport of moisture further away from its evaporative source. The water vapor in the atmosphere varies widely spatially and/or seasonally, and it is generally accepted that the residence time of water vapor in the atmosphere is approximately 8-10 days[53,54]. Nevertheless[55], proposed an optimal time for monthly integrations at a global scale for Lagrangian studies. Here, we have used the mean monthly integration time values for each region of LAR occurrence to compute the individual backward trajectories of air parcels during each LAR event.

In order to identify those regions in which there is an AMU during each LAR event, only positive values of ($E - P$) are considered. Anomalies of moisture were obtained from the difference between the individual LAR-event moisture uptake and the climatological moisture uptake for the day of the LAR for the entire period 1980–2017. Positive anomalies were calculated by adding the moisture uptake for each individual LAR event. We thus obtain those areas where the LARs gain anomalous moisture along their trajectories. For example, during the period of study (1980–2017) we report 6201 ARs events for the region 1, so we calculated 6201 fields of ($E - P$) > 0, and the same has been done for the other 19 subdomains considered here. Then, for each AR event, we calculated the anomaly of ($E - P$) > 0 and the climatology for the corresponding AR dates to verify whether these areas (where the ARs uptake on moisture) are different from the climatology. Therefore, the climatology corresponded to the same (Julian) time step but for the entire study period (again only the positive values of ($E - P$) were retained at each 6 h time step).

## Data availability
The data that support the findings of this study are available from the corresponding author upon request.

## Code availability

Code that supports the findings of this study is available upon reasonable request from the corresponding author.

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

## Acknowledgements

The present work was funded by the Spanish government within the LAGRIMA (RTI2018-095772-B-I00) project (Ministerio de Ciencia, Innovación y Universidades, Spain) and by the Xunta de Galicia, Spain, under the project Programa de Consolidación e Estructuración de Unidades de Investigación Competitivas: Grupos de Referencia Competitiva (ED431C 2017/64-GRC), which are both also funded by FEDER (European Regional Development Fund, ERDF). I.A. was financially supported by the Spanish Government (MINECO) under grant CGL2015-65141-R. J.E.-B. was financially supported by the EDB481B 2018/069 grant from the Xunta de Galicia, Spain, and the Fulbright Commission, US. We acknowledge Bin Guan for providing the AR data. This work was also supported by the project "Weather Extremes in the Euro Atlantic Region: Assessment and Impacts—WEx-Atlantic" (PTDC/CTA-MET/29233/2017) funded by Fundação para a Ciência e a Tecnologia, Portugal (FCT). Alexandre. M. Ramos was also supported by the Scientific Employment Stimulus 2017 from FCT (CEECIND/00027/2017).

## Author contributions

L.G. and R.N. designed the study; I.A. performed the research; I.A., A.M.R., and J.E.B. analyzed the data; I.A. and J.E.B. wrote the paper; L.G., R.N., A.M.R., J.E.B., R.M.T. review the paper.

## Competing interests

The authors declare no competing interests.
