## [Peer Review File · Nature Communications]

REVIEWER COMMENTS

Reviewer #1 (Remarks to the Author):

Review of "On the origin of the anomalous uptake of water vapor by landfalling Atmospheric Rivers" by Algarra et al.

The authors analyse land falling Atmospheric Rivers (ARs), intense moisture transport linked with sustained heavy precipitation, flooding and water resource impacts. Detected land-falling ARs are related to Anomalous Moisture Uptake (AMU) by the atmosphere through a novel global analysis exploiting trajectory modelling that provides further evidence for thermodynamic intensification of ARs as global temperatures rise. This appears of broad interest due to potentially wide-ranging impacts on flooding, snowpack and soil moisture. I consider this work is appropriate for Nature Communications and just have minor comments.

Comments

- 1) References - I assume just the numbers are being retained in the text but it was useful to display like this for the review
- 2) L31 - it would be useful to state the location of the WHWP (e.g. "centred on the Gulf of Mexico and the Caribbean Sea.")
- 3) L34: It is more accurate to state that extratropical cyclones are the main mode of poleward water vapour transport with ARs playing a role with reference to the discussion in the literature over whether cyclone movement or ARs within the cyclones are responsible for transporting the moisture (e.g. Dacre et al. 2019 provide evidence that the ARs represent a trail left by the cyclone when considering a cyclone-following framework)
- 4) L60: the 7%/K relates to the lower troposphere or column integrated moisture (which is mostly from the lower troposphere)
- 5) L63: as a general point, changes in ARs could be affected by changes in dynamics which alter the strength of the winds (e.g. Zhang et al. (2019) GRL doi:10.1029/2018GL079071) or alter the location and orientation of the ARs although many studies indicate thermodynamics dominate as stated (also see Lavers et al. (2013) ERL <http://doi.org/10.1088/1748-9326/8/3/034010>). It is straightforward to note that thermodynamics will intensify ARs (increasing the impact of impacts) yet the number of ARs important for impacts will relate to a threshold of moisture transport magnitude. For example, weaker ARs below the present day threshold for impacts will be promoted to impactful ARs as moisture increases. These general points could be discussed briefly.
- 6) L73: "the number of ARs detections exceeds the 10% of total days" was not completely clear to me even after looking at the methods. Does this mean number of total days in the time period considered at each point or the total number of days with land falling ARs anywhere in the world?
- 7) L86: it was not clear what was defined as anomalous with regard to AMU. Is this just greater than average or is it just computing the average of all anomalies (whether anomalous or not). AMU could be defined more clearly here.
- 8) L140: how many false hits do AR detections give (e.g. an AR is detected but no extreme precipitation)

9) Discussion: a result highlighted in the abstract is the large role of the North Atlantic. Could this be affected by decadal variability unrelated to greenhouse gas forced warming (e.g. aerosol or ocean variability e.g. Watanabe & Tatebe (2019) *Clim Dyn* <https://doi.org/10.1007/s00382-019-04811-3>). On the other hand the mid-north Atlantic ocean is cooling relative to the globe (warming hole or "cold blob") and this could have altered weather patterns. What is the global result when excluding the north Atlantic? Some further interpretation of this result and discussion of the implications would be beneficial.

10) Figure 3: I did not understand the units in Figure 3b-c. The unit mm/day is independent of area so should be averaged not summed which would be double counting and does not provide a fair comparison between grid point, latitudinal sum or global average.

11) Have methods for detecting ARs been compared and are the results sensitive to the methodology?

Reviewer #2 (Remarks to the Author):

SUMMARY

This is a clearly written, well-motivated and well-executed global study of the moisture sources for land-falling atmospheric rivers (LARs) and their evolution over nearly four recent decades. The results are new and timely as well as relevant, particularly, as the authors point out, to explaining trends in extreme precipitation from LARs. Precipitation, however, was not considered, and neither was integrated vapor transport (IVT), at least not directly, associated with wetter LARs. Below are comments and suggestions that will hopefully help improve the presentation and interpretation of results as well as the relevance/strength of the paper.

GENERAL COMMENTS

1. If this work is a step towards understanding trends in precipitation extremes, then why not actually relate trends in anomalous moisture uptake (AMU) to precipitation trends via changes in IVT? Will you actually see changes in IVT and precipitation that would be consistent with the changes in AMU detected here?
2. AMU estimates depend on modeled freshwater surface fluxes (e-p). How well is evaporation actually modeled? Is evaporation modeled better/worse than precipitation, which is certainly far from perfect? It would be useful to cite or show a validation of this measure with observations (if available) and at least include a discussion of how realistic the (e-p) modeling and the resultant AMU estimation are.

SPECIFIC COMMENTS

1. Line 42: "...certain mid latitudinal regions..." would be more accurate if it read "...certain mainly coastal mid-latitude regions..."
2. Line 45: "...and occasionally defining the end of a drought period..." – it might be good to also mention that the lack of ARs often defines drought periods, which is certainly true in the California, and is consistent with the work of Dettinger et al.
3. Line 47: "...west coast of the U.S...." => "...Western U.S...."

4. Lines 92-96: It would be useful to discuss in a little more detail which regions dominate the global AMU trend. It seems that the dominant regions are 5, 8, 13. Notably, the west coast of North America has not experienced observed increasing land-falling IVT or precipitation trends over a much longer time period (Gershunov et al. 2017, 2019) and this is consistent with the lack of AMU trends seen here in the relevant regions (Figures S6, panels b and c).

5. Lines 139-140: "...LARs...their presence can be considered an extreme precipitation event...". I doubt that presence of LARs is a sufficient condition for extreme precipitation. Doesn't extreme precipitation require the interaction of LARs with topography?

6. Lines 142-146: Shouldn't we see an increase in IVT and IWV at landfall and an increase of orographic precipitation? Checking on this would be straightforward, would provide independent verification of the AMU trends and a direct connection to floods, water resources and other societal impacts. That would greatly strengthen and benefit this paper.

7. Lines 147-157: This is a very useful discussion.

8. Lines 185-187: Regarding the exclusion of region 4, Rutz et al. (2014, 2015) specifically identified ARs landfalling at the coast of Baja California Norte as being able to penetrate inland into Arizona and New Mexico.

9. Lines 194-196: With respect to the choice of regions 1-4, in general, I realize that its hard or even impossible to make perfect choices in term of delineating relevant regions in this type of study, but the Southern California coast is very active in terms of strong LARs that are much more closely associated with southern region 3 than with LARs in Southern Baja California, which probably dominate the AMU pattern shown in Figure S2d and likely display a different seasonality than LARs in southern region 3 and northern region 4. Northern California is actually the target for the strongest ARs focused on December (Gershunov et al. 2017), and these are more associated with Southern California and to some extent with Northern Baja California LARs, while Southern Baja probably displays a very different seasonality and may be associated mainly with moisture plumes associated with intact and decaying tropical cyclones. I'm not suggesting a new definition of regions, but recognition of the importance of region 4 and its heterogeneity with respect to LARs flavors and behaviors is in order.

10. Line 212: "specify" => "specific"

I think this is excellent work and i hope my comments are useful.

Alexander Gershunov

We thank the reviewers for their kind comments that helped to improve the manuscript substantially. We have edited the manuscript accordingly. Please find below the reply to the specific comments.

Reviewer 1

Review of "On the origin of the anomalous uptake of water vapor by landfalling Atmospheric Rivers" by Algarra et al.

The authors analyse land falling Atmospheric Rivers (ARs), intense moisture transport linked with sustained heavy precipitation, flooding and water resource impacts. Detected land-falling ARs are related to Anomalous Moisture Uptake (AMU) by the atmosphere through a novel global analysis exploiting trajectory modelling that provides further evidence for thermodynamic intensification of ARs as global temperatures rise. This appears of broad interest due to potentially wide-ranging impacts on flooding, snowpack and soil moisture. I consider this work is appropriate for Nature Communications and just have minor comments.

We are grateful to the reviewer for the very positive valuation of our manuscript

Comments

1) References - I assume just the numbers are being retained in the text but it was useful to display like this for the review

Yes, the reviewer is right. The final version of the manuscript will only show the numbers, but in view of this review, we thought it appropriate to show the full version of the citations, to facilitate the reviewer's work.

2) L31 - it would be useful to state the location of the WHWP (e.g. "centred on the Gulf of Mexico and the Caribbean Sea.")

The following sentences has been added to the abstract:

"Our results also reveal generalized significant increases in AMU at the regional scale and an asymmetric supply of oceanic moisture, in which the maximum values are located over the region known as the Western Hemisphere Warm Pool (WHWP) centred on the Gulf of Mexico and the Caribbean Sea."

3) L34: It is more accurate to state that extratropical cyclones are the main mode of poleward water vapour transport with ARs playing a role with reference to the discussion in the literature over whether cyclone movement or ARs within the cyclones are responsible for transporting the moisture (e.g. Dacre et al. 2019 provide evidence that the ARs represent a trail left by the cyclone when considering a cyclone-following framework)

We are aware that there is some controversy in this regard. We would prefer to avoid addressing it explicitly in the introduction. However, in response to the reviewer's concern, the beginning of the introduction has been rewritten, using this formula, which we consider neutral and more appropriate:

"Atmospheric Rivers (ARs) are narrow regions through which large amounts of moisture are transported towards midlatitudes (e.g., 1, Gimeno et al., 2016) usually associated with the pre-frontal region of extratropical cyclones."

4) L60: the 7%/K relates to the lower troposphere or column integrated moisture (which is mostly from the lower troposphere)

We consider the reviewer's clarification. Nevertheless, it is necessary to point out that the rate of increase of the water vapor content varies widely with the height in the atmosphere. So, we refer to the near surface or at low levels of the troposphere. In the upper troposphere where the air is colder, the rate can be double. e.g. Allan (2012). We have modified the text as follows:

"The water-holding capacity of the atmosphere increases about 7% per Kelvin at the lower troposphere and for the column integrated moisture, which is mostly concentrated in the lower troposphere."

5) L63: as a general point, changes in ARs could be affected by changes in dynamics which alter the strength of the winds (e.g. Zhang et al. (2019) GRL doi:10.1029/2018GL079071) or alter the location and orientation of the ARs although many studies indicate thermodynamics dominate as stated (also see Lavers et al. (2013) ERL <http://doi.org/10.1088/1748-9326/8/3/034010>). It is straightforward to note that thermodynamics will intensify ARs (increasing the impact of impacts) yet the number of ARs important for impacts will relate to a threshold of moisture transport magnitude. For example, weaker ARs below the present day threshold for impacts will be promoted to impactful ARs as moisture increases. These general points could be discussed briefly.

We appreciate these general points that have been incorporated (together with the previous question) into the manuscript as follows:

"The water-holding capacity of the atmosphere increases about 7% per kelvin at lower troposphere or column integrated moisture which is mostly concentrated in the lower troposphere (21, O'Gorman and Muller, 2010; 22, Allan, 2012), leading to an intensification of extreme precipitation events at similar rates. We cannot disregard that under climate change, ARs may be affected by changes in dynamics that could alter the strength of the winds (e.g. 23, Zhang et al., 2019) or increase the anticyclone activity (e.g., 24, Sousa et al., 2020). However, it is generally accepted that the thermodynamically driven component dominates (25, Lavers et al., 2013; 26, Payne et al., 2020). In this context, as the amount of moisture in the atmosphere increases, so does the moisture transport. Therefore, weak ARs are bound to grow substantially in future warmer climates achieving more often the magnitude of extreme events, with a potential for greater impact for humans, ecosystems and build-up areas. Simulations of climate warming have shown more intense and frequent ARs (25, Lavers et al., 2013; 27, Ramos et al., 2016; 19, Espinoza et al., 2018; 28, Massoud et al., 2019; 29, Tan et al., 2020; 30, Kamae et al., 2019), which could lead to higher total rainfall and flooding in mid-latitude land masses."

24. Sousa, P. M. et al. North Atlantic integrated water vapor transport-from 850 to 2100 CE: Impacts on Western European rainfall. *J. Clim.* 33, 263–279 (2020).

26. Payne, A. E. et al. Responses and impacts of atmospheric rivers to climate change. *Nat. Rev. Earth Environ.* 1, 143-157 (2020).

6) L73: "the number of ARs detections exceeds the 10% of total days" was not completely clear to me even after looking at the methods. Does this mean number of total days in the time period considered at each point or the total number of days with land falling ARs anywhere in the world?

We consider as main regions of LAR those regions that have more than 10% of days of LARs for the study period, 38 years (1980-2017). All those regions that exceed this considered threshold, that is, they have more

than 1388 days of LAR (for the study period, 1980-2017) are considered for subsequent study. Following the reviewer's recommendation, we clarify this section and add the following to the text:

“For the worldwide coastline, and for the period 1980-2017, we first identified the areas of maximum occurrence of LARs (Fig. 1) as those in which the number of ARs detections exceeds the 10% of total days in the time period considered (see Methods).”

7) L86: it was not clear what was defined as anomalous with regard to AMU. Is this just greater than average or is it just computing the average of all anomalies (whether anomalous or not). AMU could be defined more clearly here.

We calculated the uptake of moisture for all individual ARs at all-time steps and saving only positive values of $(E - P)$ every 6 h during the optimal integration times for Lagrangian studies of atmospheric moisture sources and sinks (Nieto and Gimeno, 2019). For example, throughout the period of study (1980-2017) we report 6201 AR events for the region 01, so we calculated 6201 fields of $(E - P) > 0$, and the same for the other 19 subdomains considered for the moisture analysis. Then, for each AR event, we calculated the anomaly of $(E - P) > 0$ and the climatology for the corresponding AR dates, to verify whether these areas (where the ARs uptake on moisture) are different from the climatology. Therefore, the climatology corresponded to the same (Julian) time step but for the entire period (again only positive values of $(E - P)$ were retained at each 6 h time step. We agree with the reviewer in this context the word anomalous may be confusing. So, we have rewritten the sentence as follows:

“The anomalous moisture uptake (herein denoted AMU) for each region over the LARs was identified as those areas where moisture uptake was intensified during each LAR event using the Lagrangian model FLEXPART v9.0.”

Moreover, in the methods we have added the example mentioned above:

“For example, during the period of study (1980-2017) we report 6201 ARs events for the region 01, so we calculated 6201 fields of $(E - P) > 0$, and the same has been done for the other 19 subdomains considered here. Then, for each AR event, we calculated the anomaly of $(E - P) > 0$ and the climatology for the corresponding AR dates to verify whether these areas (where the ARs uptake on moisture) are different from the climatology. Therefore, the climatology corresponded to the same (Julian) time step but for the entire study period (again only the positive values of $(E - P)$ were retained at each 6 h time step).”

Nieto, R. & Gimeno, L. A database of optimal integration times for Lagrangian studies of atmospheric moisture sources and sinks. Sci. data 6, 59 (2019).

8) L140: how many false hits do AR detections give (e.g. an AR is detected but no extreme precipitation)

False AR detections will depend on the intrinsic reliability of the database used. In our case, we used the AR dataset developed by Bin Guan with ERA-Interim reanalysis data available for the period 1980-2017. This database lists all AR events that took place in that period, regardless of their associated impact. This database has been extensively validated (from this event vs non-event perspective) with different datasets and used in a large number of publications which guarantee its reliability (more information in Guan and Waliser (2015).

In relation to the reviewer's question, it is important to remember that most atmospheric rivers do not produce extreme precipitation. Although a large fraction of the top 10 or top 20 extreme precipitation events are indeed

connected with ARs in regions such as Iberia, UK or California, it is also necessary to stress that the vast majority of detected ARs do not cause extreme precipitation. Likewise, a very large majority of moderately intense precipitation events in these regions are also not driven by an AR (see e.g. for Iberia in *Ramos et al, 2015*). Moreover, most of ARs are considered beneficial and even necessary for the maintenance of the hydrological cycle (*Ralph et al. 2019*). In this sense, a detection cannot be considered a false positive just because it has not caused extreme precipitation. For an atmospheric river to produce an intense precipitation, a number of conditions have to be met, and they are not always met. In our case, the development of a 38-year-old climatology and the fact that we focus on regions of maximum activity of ARs minimize the impact of particular events of ARs on an individual analysis of the precipitation. Although it is true that the regions of AMU are calculated for each AR event, the climatology period used in this analysis minimizes the particular effect of individual ARs, buffering the possible effect of potential false positives. Moreover, it is necessary to stress that the AR database used here (as most AR databases) identifies AR events without regarding the corresponding associated precipitation. Therefore, returning to the main point raised by the reviewer's comment, there are no false ARs events in the sense that there is no need for each individual AR be associated to extreme precipitation, the only purpose of the AR database is simply to identify those events that fulfill the AR criteria.

Guan, B. & Waliser, D. E. Detection of atmospheric rivers: Evaluation and application of an algorithm for global studies. J. Geophys. Res. 120, 12,514-12,535 (2015).

Ramos, A. M., R. M. Trigo, M. L. R. Liberato, and R. Tomé, 2015: Daily Precipitation Extreme Events in the Iberian Peninsula and Its Association with Atmospheric Rivers. J. Hydrometeor., 16, 579–597, <https://doi.org/10.1175/JHM-D-14-0103.1>.

Ralph, F. M., J. J. Rutz, J. M. Cordeira, M. Dettinger, M. Anderson, D. Reynolds, L. J. Schick, and C. Smallcomb, 2019: A Scale to Characterize the Strength and Impacts of Atmospheric Rivers. Bull. Amer. Meteor. Soc., 100, 269–289, <https://doi.org/10.1175/BAMS-D-18-0023.1>.

9) Discussion: a result highlighted in the abstract is the large role of the North Atlantic. Could this be affected by decadal variability unrelated to greenhouse gas forced warming (e.g. aerosol or ocean variability e.g. Watanabe & Tatebe (2019) Clim Dyn <https://doi.org/10.1007/s00382-019-04811-3>). On the other hand the mid-north Atlantic ocean is cooling relative to the globe (warming hole or "cold blob") and this could have altered weather patterns. What is the global result when excluding the north Atlantic? Some further interpretation of this result and discussion of the implications would be beneficial.

The WHWP region appears as a particularly relevant region in terms of AMU on a global scale. Its appearance is motivated by the high activity of ARs in the North Atlantic basin that make them landfall on the west coast of Europe (regions 11, 12 and 13), the northern regions (9 and 10) and the east coast of North America (regions 6, 7 and 8). Following the reviewer's consideration, we removed the AMU regions from the North Atlantic basin from our overall results to assess its significance. The results are shown in the figure attached below (**Fig. S8**) which is now included in the supplementary material.

On the one hand, **Fig. S8A** and **S8B** represent the zonal average and annual variability excluding AMU in the North Atlantic basin from the results. **Figs. S8C** and **S8D** represent the zonal average and annual variability for ARs that develop only in the North Atlantic. From a quantitative point of view, the AMU for the ARs that develop over the North Atlantic basin accounts for approximately half of AMU, which shows the importance of this region on a global scale (**Figs. S8A and S8B**). In **Fig. S8A** we show the zonal average excluding the AMU from the ARs that develop over the North Atlantic, its elimination involves a change in the behavior of the latitudinal distribution, in which the AMU in the Northern hemisphere is surveyed by the southern hemisphere in importance. In **Fig. S8B** we show the annual variability on a global scale of the AMU which remains significant (Mann-Kendall test) despite excluding the results from the Atlantic Ocean. This result is in line with

the significant positive trends obtained over the different subdomains which causes the global trend to present a similar significant increase.

On the other hand, if we analyze the AMU of the ARs that develop only in the North Atlantic basin: a strong contribution is observed in the northern hemisphere (**Fig. S8C**). Furthermore, the results show a significant increase in AMU when its annual variability is analyzed (**Fig. S8D**).

Figure S8. AMU analysis. Zonal average of AMU and its interannual variability (shaded grey area) and worldwide annual variability of AMU. (A) and (B) excluding the AMU from North Atlantic Basin. (C) and (D) only for the AMU from North Atlantic Basin. The red lines define the region between the Tropics of Cancer (23.43°N) and Capricorn (23.43°S). The green line shows the corresponding linear regression. The trend was calculated using Mann-Kendall test which is significant in (B) and (D) (95% level of significance).

We appreciate the reviewer's consideration and we would like to add to the discussion of the results the following paragraph:

“Our results show the importance of the WHWP region in the AMU for the ARs that develop in the North Atlantic basin. The AMU from the North Atlantic accounts for almost half of the global AMU (Fig. S8a-c). When we analyze the annual variability either solely from WHWP or excluding this region from the global analysis, the AMU trend continues to increase significantly (Mann-Kendall test) (Fig. S8b-d). Our results are in line with the observed changes in the SST of the North Atlantic basin, which have revealed a warming trend associated with the North Atlantic multidecadal variability (AMV). Watanabe and Tabebe, (2019) show how anthropogenic

emissions of aerosols, particularly sulphate, may be responsible for more than half of the decadal SST variability associated with AMV.”

10) Figure 3: I did not understand the units in Figure 3b-c. The unit mm/day is independent of area so should be averaged not summed which would be double counting and does not provide a fair comparison between grid point, latitudinal sum or global average.

The caption of figure 3 in the manuscript has been modified:

Figure 3. Anomalous moisture uptake of the landfalling AR events (LARs) from 1980-2017. (A) Annual sum of positive anomalies of moisture uptake for AR detections over the main landfalling regions. The red lines define the region between the Tropics of Cancer (23.43°N) and Capricorn (23.43°S). The anomalies are calculated over the period 1980-2017. **(B)** Zonal average of AMU (blue line) and its interannual variability (shaded grey area) **(C)** Global annual variability of AMU. The green line shows the corresponding linear regression.

11) Have methods for detecting ARs been compared and are the results sensitive to the methodology?

The methodology used in our work uses the ARs database developed in *Guan and Waliser (2015)* which is widely used in AR studies and was extensively validated by different Reanalyses datasets (ERA-I, MERRA2, NCEP-NCAR) as well as other datasets (e.g. satellite, manual classification, etc), including the following examples:

- 94% agreement in detected AR landfall dates along the west coast of North America compared to the AR catalog in *Neiman et al. (2008)* based on satellite observation of integrated water vapor (IWV) and manual AR identification.
- Exact matching of mean AR duration at Bodega Bay, California compared to the result in *Ralph et al. (2013)* based on hourly observations of IWV at the Bodega Bay AR Observatory and an independent procedure for AR identification.

- 89% and 100% agreement, respectively, in AR landfall dates compared to AR detection independently conducted for Britain (*Lavers et al. 2011*) and East Antarctica (*Gorodetskaya et al. 2014*) using various reanalyses.
- Only 3% mean difference in total IVT across AR width compared to dropsonde measurements of 21 ARs in northeastern Pacific (*Guan et al. 2018*).

In particular, the regions of maximum occurrence of ARs detected in our work are analyzed with wintertime anomalies of MSLP to verify that these regions are linked to baroclinic systems. Since the AR database has an original resolution of 1.5, the detection of AR events was carried out at different spatial resolutions (1.5, 3, 6 and 12°). Finally, the accumulation of events in the highest resolution was chosen, seeking to obtain a balance between the correct identification of regions of influence of ARs (documented in the current scientific literature) with computational calculation efficiency. The aim of detection of ARs is none other than to identify the main regions of influence of ARs on a global scale for subsequent analysis in terms of AMU. In total, 24 regions of ARs were identified, after verifying the wintertime anomalies of MSLP, 4 of them were discarded, and finally we analyzed the remaining 20 regions in the subsequent study of AMU. All regions reported here are in line with previous regional and global ARs studies.

Since the detection of AR's itself is not the main objective of this paper, we have not considered appropriate to complicate the methodology by making use of different AR detection schemes/databases. Instead, we have made use of a database that is widely consolidated in the literature, and which we consider, to have yielded very positive results in previous studies. For example, Ramos et al. (2016) carried out a detection of ARs events for the west coast of Europe different from Guan and Walliser (2015), but the results from (E – P) are similar.

-
- Guan, B. & Waliser, D. E. Detection of atmospheric rivers: Evaluation and application of an algorithm for global studies. J. Geophys. Res. 120, 12,514-12,535 (2015).*
- Neiman, P. J., F. M. Ralph, G. A. Wick, J. D. Lundquist, and M. D. Dettinger. Meteorological Characteristics and Overland Precipitation Impacts of Atmospheric Rivers Affecting the West Coast of North America Based on Eight Years of SSM/I Satellite Observations. J. Hydrometeor., 9, 22–47 (2008)*
- Ralph, F. M., T. Coleman, P. J. Neiman, R. J. Zamora, and M. D. Dettinger. Observed Impacts of Duration and Seasonality of Atmospheric-River Landfalls on Soil Moisture and Runoff in Coastal Northern California. J. Hydrometeor., 14, 443–459 (2013)*
- Lavers, D. A., Allan, R. P., Wood, E. F., Villarini, G., Brayshaw, D. J., and Wade, A. J. Winter floods in Britain are connected to atmospheric rivers, Geophys. Res. Lett., 38, L23803 (2011)*
- Gorodetskaya, I. V., Tsukernik, M., Claes, K., Ralph, M. F., Neff, W. D., and M. Van Lipzig, N. P. The role of atmospheric rivers in anomalous snow accumulation in East Antarctica, Geophys. Res. Lett., 41, 6199– 6206 (2014)*
- Guan, B., D. E. Waliser, and F. M. Ralph. An Intercomparison between Reanalysis and Dropsonde Observations of the Total Water Vapor Transport in Individual Atmospheric Rivers. J. Hydrometeor., 19, 321–33 (2018)*
- Ramos, A. M. et al. Atmospheric rivers moisture sources from a Lagrangian perspective. Earth Syst. Dyn. 7, 371–384 (2016).*

Reviewer #2 (Remarks to the Author):

SUMMARY

This is a clearly written, well-motivated and well-executed global study of the moisture sources for land-falling atmospheric rivers (LARs) and their evolution over nearly four recent decades. The results are new and timely as well as relevant, particularly, as the authors point out, to explaining trends in extreme precipitation from LARs. Precipitation, however, was not considered, and neither was integrated vapor transport (IVT), at least not directly, associated with wetter LARs. Below are comments and suggestions that will hopefully help improve the presentation and interpretation of results as well as the relevance/strength of the paper.

GENERAL COMMENTS

1. If this work is a step towards understanding trends in precipitation extremes, then why not actually relate trends in anomalous moisture uptake (AMU) to precipitation trends via changes in IVT? Will you actually see changes in IVT and precipitation that would be consistent with the changes in AMU detected here?

Following the reviewer's suggestion, changes in IVT have been calculated following a methodology analogous to the calculation of anomalous moisture uptake (AMU). We have included two additional figures below that summarize the results obtained. **Fig. R1** represents the IVT anomalies for the 20 subdomains and **Fig. S9** represents the trend of global IVT anomalies. When the IVT is jointly analyzed with the AMU for the different subdomains, there is a high level of agreement in the trends of increase. However, only 3/20 regions show a significant increase in IVT anomalies compared to 14/20 regions that show a significant increase of AMU. Furthermore, two subdomains show an inverse trend when we compare anomalies of IVT and AMU: On the one hand, the Region 14 (western coast of South Africa) shows a decreasing trend in IVT whereas when we analyze the AMU it shows a trend towards a slight increase (not significant). On the other hand, region 19 (southwestern coast of Australia) presents a decrease in IVT (not significant). Nevertheless, when we study the AMU this region presents a significant increase.

The calculation of IVT anomalies is subject to the 90th AMU percentile, so regional IVT results are conditioned to the area used for its calculation. However, when we analyze the global anomalies of IVT, the results show a significantly increasing trend just as when we analyze the trend in AMU (**Fig. S9**). So, AMU anomalies are supported by IVT anomalies on a global scale. It should be noted that methodologies analogous to the calculation of AMU were successfully applied in various ARs works; for example in *Ramos et al. (2016)*, where they calculated the AMU regions for AR events reaching the western coast of Europe. The results obtained in Ramos' work are similar to those obtained here, where we identified three subdomains for the west coast of Europa with maximum AR activity and where the AMU of ARs comes mainly from the subtropical Atlantic. Another study that also supports the use of AMU, was the one conducted by *Vazquez et al. (2018)* where they study the AMU regions for the ARs that reach the Arctic system.

We included **Fig. S9** (Global annual variability of IVT anomalies) in the supplementary material, and we also added the following text to the discussion section:

Analogously to the calculation of AMU, we evaluate the regional and global trend of IVT, which is similar to the trend of AMU in some regions of maximum activity of ARs and shows a significant increase on a global scale (Fig. S9).

Figure R1. Annual variability of IVT anomalies for each LAR region from 1980 to 2017. The red line shows the corresponding linear regression. Box (top left/bottom left) shows the number of each LAR region in Fig. 1. The asterisk indicates a significant trend. Trends were calculated using Mann-Kendall test (95%). This figure was not introduced in the manuscript neither in the supplementary material.

Figure S9. Global annual variability of IVT anomalies. The green line shows the corresponding linear regression. The trend was calculated using Mann-Kendall test which is significant (95% level of significance).

M. Vázquez, I. Algarra, J. Eiras-Barca, A.M. Ramos, R. Nieto, L. Gimeno. *Atmospheric Rivers over the Arctic: Lagrangian Characterisation of Their Moisture Sources*, *Water*, 11(1), 41. Pages 1-14 (2019)

A.M. Ramos, R. Nieto, R. Tomé, L. Gimeno, R.M. Trigo, M.L.R. Liberato, D.A. Lavers. *Atmospheric rivers moisture sources from a Lagrangian perspective*, *Earth System Dynamics*, 7, 371-384 (2016)

2. AMU estimates depend on modeled freshwater surface fluxes (e-p). How well is evaporation actually modeled? Is evaporation modeled better/worse than precipitation, which is certainly far from perfect? It would be useful to cite or show a validation of this measure with observations (if available) and at least include a discussion of how realistic the (e-p) modeling and the resultant AMU estimation are.

The methodology does not allow the calculation of evaporation or precipitation separately. This is due to the intrinsic nature of the FLEXPART model. However, a validation of the available model can be found in *Stohl and James (2004; 2005)* cited in the present article. In *Stohl and James (2005)*, a validation of the E-P estimation for various river basins and main oceanic regions is performed, obtaining a good degree of agreement with both freshwater discharge data for river basins and with net water flow data for ocean basins. Their results can be favorably comparable with evaporation data from other related studies such as Dirmeyer and Brubaker (1999) and Brubaker et al. (2001). These validations turn FLEXPART into a robust methodology in the diagnosis of (E – P). Moreover, the FLEXPART model is supported by an extensive list of publications that study the atmospheric branch of the hydrological cycle from a Lagrangian perspective. In addition, several publications have recently used FLEXPART for the study of moisture characteristics in the analysis of AR events, as it is the case of the previously cited articles from *Ramos et al. (2016)*; *Vázquez et al. (2019)* or *Ramos et al. (2019)*, where the sources of moisture for the ARs reaching the west coast of South Africa are correlated with IVT anomalies, which in some way supports and validates the use of the FLEXPART model. Therefore, FLEXPART is one of the main Lagrangian methods used by the scientific community for tracing water vapor in the atmosphere and diagnosing moisture sources and sinks. The model has been used successfully in an extensive list of studies looking at moisture transport. Here are just a few of papers more the highlights:

- L. Gimeno, R. Nieto, A. Drumond, R. Castillo, R.M. Trigo (2013) Influence of the intensification of the major oceanic moisture sources on continental precipitation, *Geophysical Research Letters*, 40, 1443-1450, doi:10.1002/grl.50338
- L. Gimeno, A. Drumond, R. Nieto, R.M. Trigo, A. Stohl (2010) On the origin of continental precipitation, *Geophysical Research Letters* 37, doi: 10.1029/2010GL043712
- A.M. Durán-Quesada, L. Gimeno, J.A. Amador, R. Nieto, (2010) Moisture sources for Central America: Identification of moisture sources using a Lagrangian analysis technique, *Journal of Geophysical Research*, 15, D05103, doi: 10.1029/2009JD012455
- L. Gimeno, R. Nieto, A. Drumond, A.M. Durán-Quesada, A. Stohl, H. Sodemann, R.M. Trigo (2011) A Close Look at Oceanic Sources of Continental Precipitation, *Eos*, 92(23), 193-200
- A. Drumond, L. Gimeno, R. Nieto (2011) On the contribution of the Tropical Western Hemisphere Warm Pool source of moisture to the northern hemisphere precipitation through a lagrangian approach, *Journal of Geophysical Research*, 116, D00Q04, doi:10.1029/2010JD01539
- A. Drumond, J. Marengo, T. Ambrizzi, R. Nieto, L. Moreira, L. Gimeno (2014) The role of Amazon Basin moisture on the atmospheric branch of the hydrological cycle: a Lagrangian analysis, *Hydrology and Earth System Sciences*, vol. 18, pages 2577-2598, doi:10.5194/hessd-18-2577-2598
- R. Nieto, R. Castillo, A. Drumond, L. Gimeno (2014) A catalog of moisture sources for continental climatic regions, *Water Resources Research*, 50, doi:10.1002/2013WR013901
- L. Gimeno (2014) Oceanic sources of continental precipitation, *Water Resources Research* 50, doi:10.1002/2014WR015477
- A.M. Durán-Quesada, L. Gimeno, J. Amador (2017) Role of moisture transport for Central American precipitation, *Earth System Dynamics* 8, 147-161, doi:10.5194/esd-8-147-2017
- A.M. Ramos, R. Nieto, R. Tomé, L. Gimeno, R.M. Trigo, M.L.R. Liberato, D.A. Lavers (2016) Atmospheric rivers moisture sources from a Lagrangian perspective, *Earth System Dynamics*, 7, 371-384, doi:10.5194/esd-7-371-2016
- M. Vázquez, I. Algarra, J. Eiras-Barca, A.M. Ramos, R. Nieto, L. Gimeno (2019) Atmospheric Rivers over the Arctic: Lagrangian Characterisation of Their Moisture Sources, *Water*, 2019, 11(1), 41. Pages 1-14 doi:10.3390/w11010041
- A.M. Ramos, R.C. Blamey, I. Algarra, R. Nieto, L. Gimeno, R. Tomé, C. Reason, R.M. Trigo (2019) From Amazonia to southern Africa: atmospheric moisture transport through low-level jets and atmospheric rivers, *Annals of the New York Academy of Sciences*, Vol. 1436, Issue1 Pages 217-230 DOI: 10.1111/nyas.13960

More published works can be consulted at the following link:

<https://ephyslab.uvigo.es/moisturetransport/index.php/Publications>

Stohl, A. & James, P. A Lagrangian analysis of the atmospheric branch of the global water cycle: Part 1: Method description, validation, and demonstration for the August 2002 flooding in central Europe. *J. Hydrometeorol.* 5, 656–678 (2004).

Stohl, A., Forster, C., Frank, A., Seibert, P. & Wotawa, G. Technical note: The Lagrangian particle dispersion model FLEXPART version 6.2. *Atmos. Chem. Phys.* 5, 2461–2474 (2005).+

Stohl, A. & James, P. A Lagrangian analysis of the atmospheric branch of the global water cycle. Part II: Moisture transports between earth's ocean basins and river catchments. *J. Hydrometeorol.* 6, 961–984 (2005).

SPECIFIC COMMENTS

1. Line 42: "...certain mid latitudinal regions..." would be more accurate if it read "...certain mainly coastal mid-latitude regions..."

The cited sentence has been rewritten following the reviewer's recommendation:

“Most of extreme precipitation and flood events are associated with landfalling AR events for certain coastal mid-latitude regions, especially when subjected to orographic lift over mountainous topography (5, Dettinger et al., 2011).”

2. Line 45: “...and occasionally defining the end of a drought period...” – it might be good to also mention that the lack of ARs often defines drought periods, which is certainly true in the California, and is consistent with the work of Dettinger et al.

We take into account the reviewer's appreciation and add their consideration to the manuscript in line with the work of Dettinger, 2013:

“They have been linked to a wide range of socio-economic impacts, affecting the severity and frequency of flooding, and occasionally defining the end of a drought period. Additionally, it has been shown that the lack of ARs is often correlated with drought periods (e.g., 6, Dettinger, 2013).”

3. Line 47: “...west coast of the U.S....” => “...Western U.S....”

The sentence has been rewritten following the reviewer's recommendation.

4. Lines 92-96: It would be useful to discuss in a little more detail which regions dominate the global AMU trend. It seems that the dominant regions are 5, 8, 13. Notably, the west coast of North America has not experienced observed increasing land-falling IVT or precipitation trends over a much longer time period (Gershunov et al. 2017, 2019) and this is consistent with the lack of AMU trends seen here in the relevant regions (Figures S6, panels b and c).

We add to the results a brief comment following the reviewer's consideration. In addition, we have added a paragraph in the discussion section where we discuss the importance of the WHWP region, which dominates the AMU.

“It is necessary to emphasize that the region of the Western U.S. does not show a significant trend, which is in line with the work of Gershunov et al. (2017; 2019) where no increase in IVT or precipitation trends have been found over a large period of time (Figs. s6, b-c).”

Following reviewer's 1 suggestion, we also included this paragraph in the discussion:

“Our results show the importance of the WHWP region in the AMU for the ARs that develop in the North Atlantic basin. The AMU from the North Atlantic accounts for almost half of the global AMU (Fig. s8a-c). When we analyze the annual variability either solely from WHWP or excluding this region from the global analysis, the AMU trend continues to increase significantly (Mann-Kendall test) (Fig. s8b-d). Our results are in line with the changes in the SST of the North Atlantic basin, which have revealed a warming trend associated with the North Atlantic multidecadal variability (AMV). 39, Watanabe and Tatebe, (2019) show how anthropogenic emissions of aerosols, particularly sulphate, may be responsible for more than half of the decadal SST variability associated with AMV”.

5. Lines 139-140: "...LARs...their presence can be considered an extreme precipitation event...". I doubt that presence of LARs is a sufficient condition for extreme precipitation. Doesn't extreme precipitation require the interaction of LARs with topography?

We take into account the reviewer's appreciation and add their consideration. The paragraph has been rewritten as follows:

"Given the critical role of LARs for rainfall and flooding in many regions, their presence *may be associated with extreme precipitation, particularly when uplift, either orographic or thermodynamic (via warm conveyor belt), play a significant role* (8, Lavers et al., 2011; 9, Lavers and Villarini, 2013; 11, Ramos et al., 2015; 10, Eiras-Barca et al., 2016). Although a large fraction of the top 20 extreme precipitation events are indeed connected with ARs in regions such as the Iberian Peninsula, it is also necessary to stress that many of the detected ARs do not cause extreme precipitation (11, Ramos et al., 2015).

Eiras-Barca, J., Brands, S., & MiguezMacho, G. (2016). Seasonal variations in North Atlantic atmospheric river activity and associations with anomalous precipitation over the Iberian Atlantic Margin. *Journal of Geophysical Research: Atmospheres*, 121(2), 931-948.

Ramos, A. M., R. M. Trigo, M. L. R. Liberato, and R. Tomé (2015). Daily Precipitation Extreme Events in the Iberian Peninsula and Its Association with Atmospheric Rivers. *J. Hydrometeorol.*, 16, 579–597

6. Lines 142-146: Shouldn't we see an increase in IVT and IWV at landfall and an increase of orographic precipitation? Checking on this would be straightforward, would provide independent verification of the AMU trends and a direct connection to floods, water resources and other societal impacts. That would greatly strengthen and benefit this paper.

We calculated the trend of orographic precipitation for the western coast of U.S. (region 03). Nevertheless, it should be noted that the regions of maximum occurrence of ARs selected for the AMU study are quite wide, therefore complicating significantly the analysis in terms of orographic precipitation. For example, for the western coast of U.S., an AR event in the north of this region may not involve precipitation in the south and vice versa. Considering the above, we show below the trend of anomalous orographic precipitation (**Fig. R2a** calculated analogously to the AMU, from maximum precipitation.-data: ERA-Interim) in region 03, western coast of U.S. (region widely documented in terms of orographic precipitation). Although the positive trend in orographic precipitation is clearly observed, the large inter-annual variability implies that is not statistically significant. Nevertheless this positive trend is compatible with the AMU and IVT trends shown for the same region and period (**Fig. R2b-c**).

Figure R2. (A) Maximum precipitation anomaly (mm) for the region 03 (western coast of U.S.). (B) Anomalous moisture uptake (AMU, mm) for the region 03 and (C) Anomalous IVT for the region (03). The red line shows the corresponding linear regression.

7. Lines 147-157: This is a very useful discussion.

We appreciate the reviewer's comment.

8. Lines 185-187: Regarding the exclusion of region 4, Rutz et al. (2014, 2015) specifically identified ARs landfalling at the coast of Baja California Norte as being able to penetrate inland into Arizona and New Mexico.

Particularly, the region 4 appears as a region of great influence of ARs. In fact we refer to Rutz's works (Rutz et al., 2014; 2015) in the methods section. However, as we explained in the methods section, when we analyze the MSLP anomalies for this particular region we did not find a relatively deep negative anomaly during winter that is associated with low pressures. This may be due to the influence of tropical storms and monsoon circulations in region 4, just as it happens in regions 15 and 16 and which were discarded for the purposes of moisture transport analysis.

Rutz, J. J., James Steenburgh, W. & Martin Ralph, F. Climatological characteristics of atmospheric rivers and their inland penetration over the western united states. Mon. Weather Rev. 142, 905–921 (2014).

Rutz, J. J., James Steenburgh, W. & Martin Ralph, F. The inland penetration of atmospheric rivers over western North America: A Lagrangian analysis. Mon. Weather Rev. 143, 1924–1944 (2015).

9. Lines 194-196: With respect to the choice of regions 1-4, in general, I realize that its hard or even impossible to make perfect choices in term of delineating relevant regions in this type of study, but the Southern California coast is very active in terms of strong LARs that are much more closely associated with southern region 3 than with LARs in Southern Baja California, which probably dominate the AMU pattern shown in Figure S2d and likely display a different seasonality than LARs in southern region 3 and northern region 4. Northern California is actually the target for the strongest ARs focused on December (Gershunov et al. 2017), and these are more associated with Southern California and to some extent with Northern Baja California LARs, while Southern Baja probably displays a very different seasonality and may be associated mainly with moisture plumes associated with intact and decaying tropical cyclones. I'm not suggesting a new definition of regions, but recognition of the importance of region 4 and its heterogeneity with respect to LARs flavors and behaviors is in order.

Rutz et al., (2014; 2015) defines regions objectively based on the penetration of ARs into the west coast of North America. The third region that they consider ranges from the border with Mexico along Baja California where ARs that penetrate land usually curve towards the southern United States. This subdivision is very useful from the rainfall assessment perspective. However, Rutz's AR ground penetration criterion is not relevant in terms of moisture sources when conducting the subdivision. In our particular case, we have considered that it is better to adopt a criterion of geographic and latitudinal orientation, more in line with the work of, for example, *Payne and Magnusdottir, (2015)* or *Kim et al. (2019)* which we believe is more suitable to assess the diversity of ARs and their associated sources of moisture given the extensive study region such as the western coast of North America. However, we understand the importance of region 4 in terms of ARs, so we took into account the reviewer comments and we have modified the text as follows:

“Nevertheless, the Southern California coast is a very active region in terms of strong LARs, which are more closely associated with southern region 3 than LARs in southern Baja California. In fact, the strongest ARs in Northern California are mostly observed in December (34, Gershunov et al., 2017). These ARs are associated with the Southern California and to some extent with Northern Baja California LARs, while Southern Baja

displays a seasonality and a stronger marked difference, partly also due to the influence of the North American monsoon circulation.”

Rutz, J. J., James Steenburgh, W. & Martin Ralph, F. Climatological characteristics of atmospheric rivers and their inland penetration over the western united states. Mon. Weather Rev. 142, 905–921 (2014).

Rutz, J. J., James Steenburgh, W. & Martin Ralph, F. The inland penetration of atmospheric rivers over western North America: A Lagrangian analysis. Mon. Weather Rev. 143, 1924–1944 (2015).

Payne, A. E. & Magnusdottir, G. An evaluation of atmospheric rivers over the North Pacific in CMIP5 and their response to warming under RCP 8.5. J. Geophys. Res. Atmos. 120, 11,173-11,190 (2015).

Kim, H. M., Zhou, Y. & Alexander, M. A. Changes in atmospheric rivers and moisture transport over the Northeast Pacific and western North America in response to ENSO diversity. Clim. Dyn. 52, 7375–7388 (2019).

Gershunov, A., Shulgina, T., Ralph, F. M., Lavers, D. A. & Rutz, J. J. Assessing the climate-scale variability of atmospheric rivers affecting western North America. Geophys. Res. Lett. 44, 7900–7908 (2017).

10. Line 212: “specify” => “specific”

Thank you. It has been corrected on the text.

I think this is excellent work and i hope my comments are useful.

We really appreciate the reviewer’s help.

REVIEWERS' COMMENTS:

Reviewer #1 (Remarks to the Author):

The authors have responded comprehensively to my suggestions and with the consideration of some minor suggestions below I consider that the manuscript should be published.

1) Abstract: for consistency with the review comments I suggest changing "which provide the main mechanism of" to "which are an important mechanism of"

2) L77 "growth"  "grow"

3) L176 Based on the review responses I recommend adding to the caveat of only a fraction of AR events lead to extremes that extremes are also related to non-AR events.

4) L186 "high"  "strong"

5) L193 "important"  "substantial"

Reviewer #2 (Remarks to the Author):

The authors have thoughtfully and adequately addressed my comments and concerns. I find the revised paper significantly improved. I have no more requests and am happy to recommend this important paper for publication.

Alexander Gershunov